# Cardiac CT in Non-Obstructive Coronary Artery Disease (NOCAD): A Literature Review

**DOI:** 10.3390/jcm15010032

**Published:** 2025-12-20

**Authors:** Sofia Meossi, Carmen Izzo, Laura Rotondo, Giorgio Sciaramenti, Edoardo Menzato, Beatrice Dal Passo, Renè Tezze, Federica Frascaro, Elisabetta Tonet, Federico Marchini, Gianluca Campo, Rita Pavasini

**Affiliations:** Cardiology Unit, Azienda Ospedaliero-Universitaria di Ferrara, 44124 Ferrara, Italy; sofimeossi@gmail.com (S.M.); carmen.izzo2017@gmail.com (C.I.); laura.rotondo11@gmail.com (L.R.); giorgio.sciaramenti@gmail.com (G.S.); edoardo.menzato@gmail.com (E.M.); beatrice.dalpasso@edu.unife.it (B.D.P.); rene.tezze@gmail.com (R.T.); federica.frascaro92@gmail.com (F.F.); mrcfrc2@unife.it (F.M.); cmpglc@unife.it (G.C.)

**Keywords:** cardiac CT, non obstructive coronary artery disease, NOCAD, CAD

## Abstract

Non-obstructive coronary artery disease (NOCAD) encompasses a heterogeneous group of conditions in which patients present with angina, ischemia or myocardial infarction despite the absence of obstructive epicardial stenoses. This spectrum includes myocardial infarction with non-obstructive coronary arteries (MINOCA) and angina or ischemia with non-obstructive coronary arteries (ANOCA/INOCA), entities increasingly recognized as clinically significant and associated with adverse outcomes. Advances in cardiac computed tomography (CT) have expanded the diagnostic capabilities beyond the exclusion of obstructive coronary artery disease, enabling comprehensive anatomical, functional and tissue-level assessment relevant to NOCAD. CT allows precise identification of non-obstructive atherosclerosis, high-risk plaque features, myocardial bridging and structural vascular remodelling. Quantitative and qualitative characterization of plaque burden correlates with ischemic risk and provides prognostic information that complements traditional stenosis-based evaluation. Emerging CT-derived biomarkers, such as pericoronary fat attenuation index and epicardial adipose tissue metrics, offer insight into vascular inflammation and microvascular dysfunction, key mechanisms in NOCAD. Functional CT techniques, such as CT-derived fractional flow reserve and CT perfusion imaging, enable non-invasive assessment of hemodynamic significance and microvascular impairment, although their routine use is limited by methodological variability and evolving clinical evidence. Beyond coronary evaluation, CT also provides myocardial tissue characterization, detects extracardiac causes of symptoms and contributes to comprehensive differential diagnosis. Despite its strengths, cardiac CT remains limited by spatial resolution, radiation exposure and its inability to directly visualize the microcirculation. Nevertheless, ongoing technological refinement and integration of computational modelling are likely to enhance its diagnostic and prognostic role.

## 1. Introduction

Although obstructive atherosclerotic coronary artery disease (CAD) is the leading cause of ischemic heart disease in the general population, a substantial subset of individuals presenting with angina or acute coronary syndromes are found to have no obstructive coronary artery disease on angiography. Non-obstructive coronary artery disease (NOCAD) refers to a heterogeneous spectrum of clinical entities defined as anatomical stenosis less than 50% on invasive or non-invasive coronary imaging which may manifest as angina, ischemia, or myocardial infarction [1,2,3]. Advances in cardiac and coronary computed tomography (CT) technology now allow for detailed anatomical and functional characterization of coronary arteries and can guide the need for invasive coronary angiography. Cardiac CT provides high sensitivity for excluding obstructive CAD and can identify non-obstructive atherosclerosis, myocardial bridging, and other coronary abnormalities while CT-derived fractional flow reserve and CT perfusion imaging further enhance functional assessment, supporting clinical decision-making in patients with NOCAD [4,5]. This narrative review explores the potential applications and advantages of cardiac and coronary CT in the assessment of NOCAD.

## 2. Definitions

To correctly understand the pathophysiology and the application of CT in the setting of NOCAD, it is necessary to define some specific clinical scenarios (Figure 1):Myocardial Infarction with Non-Obstructive Coronary Arteries (MINOCA) can be diagnosed if the criteria of the fourth universal definition of myocardial infarction are fulfilled and obstructive coronary artery disease is ruled out by coronary angiography (no lesion ≥ 50% in a major epicardial vessel) [6,7,8]. MINOCA accounts for approximately 6–15% of all myocardial infarctions and encompasses a wide range of mechanisms, including plaque disruption, coronary artery spasm, coronary dissection, coronary embolism, type 2 myocardial infarction (MI), and Takotsubo cardiomyopathy [7,9]. Traditionally regarded as a benign entity, MINOCA is now recognized to confer a significantly increased risk of adverse cardiovascular outcomes compared with the general population [10,11].Angina with Non-Obstructive Coronary Arteries (ANOCA) and Ischemia with Non-Obstructive Coronary Arteries (INOCA): the difference between these two entities, although minimal, lies in their definition. ANOCA highlights the presence of anginal symptoms, while INOCA focuses on evidence of myocardial ischemia (such as abnormal stress testing or imaging) in the absence of obstructive CAD (<50% stenosis). The term ANOCA is generally used in cases of symptoms related to suspected myocardial ischemia that has not yet been identified or that could not be documented by instrumental imaging, while the term INOCA is used in the presence of documented ischemia, even in the absence of symptoms. Both conditions often share the same pathophysiological mechanisms, as the mismatch between blood supply and myocardial oxygen demands are primarily caused by coronary microvascular dysfunction, microvascular spasm, endothelial dysfunction, epicardial spasm, and myocardial bridging [1,2,3,12,13].

### 2.1. Pathophysiological Mechanisms of MINOCA

MINOCA encompasses multiple etiologies, requiring further evaluation when epicardial arteries are free of obstructive lesions, as management depends on the underlying cause (Table 1) [6,7]. Plaque disruption (rupture, ulceration, or calcific nodules) can lead to thrombosis, vasospasm, and myocardial necrosis, though angiography may appear normal due to spontaneous thrombolysis; intravascular imaging identifies plaque disruption in about 40% of cases [6,7,10,14]. Coronary vasospasm, spontaneous or triggered (e.g., cocaine, methamphetamine), is common, with inducible spasms observed in 27% of patients [14,15]. Coronary thrombosis and embolism may arise from hereditary or acquired thrombotic disorders or embolic sources such as valvular vegetations, cardiac tumors, or iatrogenic air emboli [7,15]. Spontaneous coronary artery dissection (SCAD) can cause infarction even with normal angiography, is linked to fibromuscular dysplasia, and is more frequent in women under 50, accounting for up to 25% of acute MI in this group [10,14,15,16]. Takotsubo syndrome represents approximately 1–2% of MINOCA cases and should be considered in the differential diagnosis [14,15].

### 2.2. Pathophysiological Mechanisms of ANOCA/INOCA

In chronic coronary syndromes, angina may result from vasospasm, microvascular dysfunction, or anatomical variants such as myocardial bridging (MB) (Table 1). Microvascular dysfunction is a key mechanism in ANOCA and INOCA, affecting 30–50% of patients with chest pain and non-obstructive coronary arteries [7,17]. It is characterized by functional abnormalities (impaired endothelial vasodilation, increased microvascular resistance, spasm) and structural changes (arteriolar remodeling, capillary rarefaction, fibrosis) [18,19]. It is particularly common in women and those with cardiovascular risk factors and is associated with heart failure with preserved ejection fraction and major adverse cardiovascular events [18,20,21]. MB, a congenital anomaly where a coronary segment runs intramyocardially, causes dynamic systolic compression and altered flow, and is closely linked to microvascular dysfunction [2,22]. Its prevalence varies by diagnostic method (2–6% on angiography, 19–22% on CT angiography) [23,24]. Although often benign, MB can induce ischemia through mechanical compression, stress-induced demand, and endothelial dysfunction, and its coexistence with microvascular dysfunction increases ischemic risk and symptom burden [2,22,25]. Clinically, MB in ANOCA/INOCA is associated with recurrent angina, hospitalizations, and reduced quality of life [26,27].

## 3. Role of CT in NOCAD

Cardiac computed tomography (CCT) is the main non-invasive technique for coronary anatomy assessment and coronary plaque evaluation which has gone through an exponential growth during the last decades due to its high diagnostic accuracy and negative predictive value [28]. Beyond the evaluation of obstructive CAD, cardiac CT also plays a critical role in the etiological assessment of NOCAD, as it allows for the identification of specific coronary and cardiac features that may aid in the differential diagnosis in this complex clinical setting (Figure 2) [29].

### 3.1. Plaque Characteristics

Beyond the detection of coronary artery anomalies, stenosis and myocardial bridges, CCT has a suite of strengths in the identification of vulnerable plaque features and total atherosclerotic plaque burden, both of which are critical in the evaluation of NOCAD [30,31]. CCT can reliably identify high-risk plaque characteristics, showing results comparable to intracoronary imaging modalities. Moreover, it has proven valuable for longitudinal monitoring of plaque dynamics, including progression, regression or stabilization, demonstrating non-inferior to IVUS [31,32]. The ability of CCT to evaluate the entire coronary vessel for atherosclerosis, as compared with only segmental approach by IVUS or OCT, is crucial to identify and predict ischemia [33,34,35]. Among the defining plaque features assessed by CT, the napkin-ring sign stands out as a marker of plaque vulnerability. This sign is characterized by a central low-attenuation area, consistent with a large necrotic core, circumscribed by a higher attenuation rim corresponding to fibrous tissue. Histopathological correlation confirms the napkin-ring sign as indicative of unstable plaques, which are prone to rupture and adverse cardiac events [36]. Another hallmark of high-risk plaques is positive remodelling, where the vessel wall expands outward at the plaque site, preserving the lumen size despite significant atheroma burden. Positive remodelling has demonstrated high sensitivity and specificity for identifying culprit lesions leading to acute coronary syndromes (ACS) and is a vital feature to assess in NOCAD patients who often lack flow-limiting stenoses [37]. Presence of spotty calcifications, defined as small, punctate calcium deposits less than 1.5 times the vessel diameter, also correlates with plaque instability and acute coronary events. These microcalcifications likely result from active inflammation and microstructural changes within the plaque [38]. In addition, low-attenuation plaques with densities below 30 Hounsfield units on CT typically represent lipid-rich necrotic cores, linked to higher risk profiles for plaque rupture and progression [39]. Together, napkin-ring sign, positive remodelling, spotty calcifications and low-attenuation plaques, are crucial for risk stratification in NOCAD and improve prognostic accuracy beyond stenosis evaluation alone (Figure 3) [38,40].

There is also evidence that patients with MINOCA or Takotsubo syndrome more frequently have coronary artery plaques, although nonobstructive, that are more often classified as high-risk plaques, compared with subjects with no coronary artery disease or with nonobstructive disease [41].

Recently, it has been shown that among high-risk plaque features, positive remodelling is more strongly correlated with the degree of perivascular inflammation and vulnerability, as it is significantly associated with higher lesion-specific pericoronary adipose tissue attenuation, a topic discussed in detail in the following section [42].

### 3.2. Inflammation of Pericoronary and Epicardial Fat

Pericoronary adipose tissue (PCAT) and epicardial adipose tissue (EAT) are metabolically active fat depots that closely interact with coronary arteries and myocardium, contributing to vascular inflammation, atherosclerosis progression, and microvascular dysfunction in both CAD and NOCAD (Figure 4) [41,43]. Cardiac CT enables non-invasive assessment of perivascular inflammation through the Pericoronary Fat Attenuation Index (pFAI), with higher values indicating increased local inflammation [44]. The CRISP CT trial demonstrated that pFAI values ≥ −70.1 HU are associated with increased all-cause and cardiac mortality [45]. pFAI may help differentiate inflammatory from non-inflammatory coronary states and elucidate mechanisms underlying MINOCA, such as vasospasm linked to coronary inflammation [43,46].

EAT, located between the myocardium and visceral pericardium, shares microcirculation with coronary arteries and exerts proinflammatory, pro-atherogenic, and pro-thrombotic effects via adipocytokine secretion [47,48,49]. Increased EAT volume and density on cardiac CT correlate with CAD severity, plaque burden, microvascular dysfunction, and adverse outcomes [47,48,49]. EAT-related inflammation likely promotes endothelial dysfunction and impaired coronary flow reserve, contributing to ischemia in NOCAD patients despite non-obstructive epicardial disease [48,50]. Elevated pFAI and EAT have been reported in microvascular angina and NOCAD, correlating with endothelial dysfunction and reduced coronary flow reserve, suggesting their role as imaging biomarkers of ischemic risk [50,51,52].

Recent evidence also links perivascular inflammation with coronary calcification, plaque vulnerability, and adverse outcomes in NOCAD, supporting the concept of inflammation-driven plaque instability even in the absence of significant luminal stenosis [53]. CT-derived inflammatory metrics may further serve as non-invasive biomarkers for monitoring therapeutic response, as longitudinal studies have shown reductions in pericoronary inflammation and plaque stabilization with high-dose statin therapy [54].

Despite promising evidence, clinical implementation of CT-derived inflammatory biomarkers is limited by technical variability, costs, and the need for specialized expertise, highlighting the necessity for further standardization and cost-effectiveness analyses. Overall, integrating CT-derived plaque features, perivascular inflammation, and functional indices offers a more comprehensive phenotyping of NOCAD (Table 2), improving risk stratification and supporting personalized management beyond stenosis severity alone.

### 3.3. Vascular Remodelling in Microvascular Dysfunction

Non-obstructive angina and ischemia in NOCAD often stem from coronary microvascular dysfunction (CMD), a condition challenging to diagnose due to the absence of epicardial stenosis [55]. CCT is at present widely used and recommended as a first-line test in the diagnostic work-up of patients with low-to-intermediate likelihood of obstructive CAD, mainly due to its high negative predictive value in ruling out the disease, which is also an essential finding for the identification of patients with CMD [5,56,57,58]. Recent advances have enabled assessment of vascular remodelling in CMD via cardiac CT by measuring the epicardial coronary artery lumen volume. Structural remodelling in CMD is characterized by a significant reduction in epicardial lumen volume, which correlates closely with increased microvascular resistance measured invasively [59]. Studies show that patients with CMD and heightened microvascular resistance exhibit a 40% smaller epicardial lumen volume than controls, indicating pathological outward vascular remodelling that is not visible as focal stenosis but contributes to impaired coronary flow. This reduced lumen volume reflects an interplay between vascular smooth muscle dysfunction and perivascular inflammation, influencing coronary flow reserve and ischemic symptoms in NOCAD [59,60]. On the other hand, functional tests can now also be applied to assess the presence of microcirculatory disorders in patients without obstructive CAD. In particular, two CCT-based functional approaches have recently been proposed for CMD detection: Fractional Flow Reserve CT (FFR-CT; HeartFlow, Redwood City, CA, USA) and CT perfusion (CTP), which will be discussed in detail in the following sections [28,55]. Thus, cardiac CT quantitative assessment of epicardial lumen volume offers a non-invasive window into microvascular health and remodelling, providing mechanistic insight into ischemia in the absence of significant coronary stenosis.

### 3.4. Cardiac CT for Myocardial and Extracardiac Assessment in NOCAD

Cardiac CT is increasingly applied not only to coronary anatomy but also to myocardial tissue characterization. Recently, late iodine enhancement (LIE) has emerged as a topic of interest: analogous to late gadolinium enhancement on cardiac magnetic resonance (CMR), LIE identifies regions of myocardial fibrosis, scarring, or infarction by delayed washout of contrast agents [61,62]. In patients with NOCAD, the presence of LIE can reveal subclinical infarction, microvascular obstruction and fibrotic remodelling that contribute to ischemic symptoms. Cardiac CT also allows the assessment of cardiac wall thinning, which typically corresponds to chronic infarcted or non-viable myocardium, reflecting adverse remodelling and impaired contractility [60,61]. CT can also assess the embolic source in the cardiac chamber, as well as the presence of both venous thrombus and right-to-left shunts (patent foramen ovale, atrial septal defect or coronary artery fistula) [63]. Moreover, cardiac CT can detect early perfusion defects and subtle regional wall motion abnormalities [64,65]. This myocardial tissue characterization provides key diagnostic and prognostic information and may guide targeted therapies in NOCAD patients with ischemia and myocardial dysfunction [5,66]. Finally, CT can aid in the differential diagnosis of NOCAD by identifying extracardiac causes. Indeed, CT can be used to clearly diagnose acute pulmonary embolism (PE) with high accuracy, where a central filling defect within a vessel surrounded by contrast material is a direct sign of PE [67]. An aortic dissection may also cause elevated troponin due to extension of the intimo-medial flap to the coronary artery or decreased myocardial flow leading to damaged myocardial cells [63].

## 4. Prognostic Classifications Derived from CCT

Several classification systems have been developed to stratify prognosis in patients with non-obstructive CAD (NOCAD) (Figure 5) [68,69]. Early dichotomous schemes (<50% stenosis vs. none) have evolved into more detailed systems, including NOCAD-RADS (0%, 1–24%, 25–49%) and the Duke prognostic index, which consider both degree and distribution of disease. The stenosis proximal involvement (SPI) classification highlights plaques in proximal coronary segments and has been shown in a Chinese cohort to independently predict adverse events, performing comparably to traditional indices [68,69]. The CT-adapted Leaman score (CT-LeSc) combines stenosis severity, plaque composition, and lesion location, identifying NOCAD patients with CT-LeSc > 5 who exhibit event-free survival similar to obstructive CAD. These tools underscore the heterogeneity of NOCAD and the value of CT-based scores for refined risk stratification and therapeutic guidance [68,69].

## 5. Functional Assessment by Cardiac CT in NOCAD

Functional evaluation using cardiac CT has increasingly complemented anatomical imaging in diagnosing and managing patients with NOCAD.

### 5.1. The Role of FFR-CT

The cardiac computed tomography-derived fractional flow reserve (FFR-CT) is an emerging non-invasive technique that plays a significant role in the assessment of NOCAD. FFR-CT uses routine CCT data combined with computational fluid dynamics according to the Navier–Stokes equations in order to simulate blood flow and pressure, providing a physiological evaluation of coronary lesions [5,28]. This involves creation of an anatomic 3D model of the entire coronary arterial system according to semiautomatic segmentation of the volumetric CCT data, as well as a physiologic model according to patient-specific inflow and outflow boundary condition assumptions, thereby theoretically mimicking the conditions of invasive FFR. Analogous to conventional FFR, this technique allows estimation of the physiologic impact of a stenosis at maximum hyperemia, with the same threshold value of 0.80 or less [70]. This method improves diagnostic accuracy by distinguishing hemodynamically significant lesions from anatomically evident but functionally irrelevant stenoses, which is particularly relevant in NOCAD where stenosis may be mild or absent, yet ischemia is present. Studies show that FFR-CT can identify subtle functional impairments in coronary circulation, influenced by vessel morphology and plaque characteristics, thereby refining risk stratification and guiding clinical management beyond traditional anatomical assessment alone [71]. Beyond its diagnostic role, the prognostic implications of FFR-CT in NOCAD are under investigation. Wang and colleagues conducted a retrospective cohort study in patients with suspected CAD and non-obstructive findings on CCT. They compared three predictive models for major adverse cardiovascular events: the first included only clinical variables, the second incorporated both clinical variables and CCT-based measures of atherosclerotic burden such as the Leiden risk score and the segment involvement score and the third added FFR-CT to these parameters. FFR-CT was found to provide independent and incremental prognostic information, and the combination of FFR-CT with the Leiden risk score further improved long-term risk prediction. These results suggest that FFR-CT, beyond its functional role, may contribute meaningfully to risk stratification in NOCAD. The prognostic significance of this approach is currently being evaluated in larger prospective trials, such as the China FFR-CT Study 2, which is investigating the relationship between FFR-CT, CCT findings and long-term outcomes in patients with non-obstructive CAD [72,73]. The ability of FFR-CT to complement anatomical information has been highlighted in studies exploring the relationship between plaque morphology and ischemia in non-obstructive CAD. Feuchtner and colleagues showed that non-obstructive lesions with high-risk plaque characteristics were associated with significantly lower lesion-specific and distal FFR-CT values compared with calcified lesions, indicating greater ischemic potential. Among individual high-risk plaque features, positive remodelling correlated most strongly with ischemia, while spotty calcification showed no independent association, and the napkin-ring sign only a modest one. Similarly, lower plaque density—suggestive of a lipid-rich component—was linked to lower FFR-CT values, reinforcing the concept that compositional vulnerability influences hemodynamic relevance. Taken together, these findings underscore the added value of combining CCT with FFR-CT, as this approach allows simultaneous assessment of plaque anatomy and its functional consequences [74].

### 5.2. The Role of Cardiac CTP

Microvascular dysfunction is one of the pathophysiological mechanisms leading to NOCAD, yet its non-invasive evaluation remains challenging. Positron emission tomography is the reference standard for quantifying myocardial blood flow, and CMR has also been validated for this purpose. Cardiac CT perfusion (CTP) offers the potential to integrate anatomical and functional information within a single examination [75,76]. CTP can be performed in static or dynamic mode. Static imaging provides a qualitative assessment by highlighting hypoattenuating myocardial regions indicative of perfusion deficits. Dynamic CTP, in contrast, generates quantitative perfusion metrics such as myocardial blood flow, blood volume, peak enhancement and time to peak. Preclinical studies have shown that CT-derived myocardial blood flow correlates strongly with microsphere and CMR measurements. In clinical cohorts, CT-derived perfusion estimates have demonstrated good agreement with rubidium-82 PET [77]. Nevertheless, despite these promising findings, CT perfusion is not yet widely used in routine practice to evaluate microvascular disease, as evidence remains limited and acquisition protocols lack standardization. Furthermore, questions persist about whether iodinated contrast agents may themselves induce vasodilation, potentially leading to overestimation of flow. Future studies aimed at protocol harmonization and large-scale validation are required, but dynamic CTP may eventually represent a viable alternative to PET for quantitative assessment of myocardial blood flow in NOCAD [78].

## 6. Limitations of Cardiac CT in NOCAD

Although cardiac CT has revolutionized non-invasive coronary imaging and provides anatomical and functional insights in NOCAD, several important limitations must be acknowledged. Cardiac CT predominantly visualizes epicardial coronary arteries and luminal stenosis but lacks direct visualization of coronary microcirculation, since microvascular abnormalities occur below the spatial resolution of CT imaging. Thus, while CT can rule out epicardial obstructive lesions, it cannot definitively diagnose microvascular ischemia or endothelial dysfunction, often requiring complementary functional or invasive testing [79,80]. Additionally, plaque characterization by CT still suffers from limited tissue resolution, especially in case of heavily calcified plaques, where blooming artifacts cause overestimation of stenosis severity and reduce diagnostic specificity, particularly in patients with high coronary calcium scores [81,82,83,84]. Moreover, cardiac CT image quality can be compromised by patient-related and technical factors: high heart rates, arrhythmias, breathing motion, beam-hardening artifacts from dense calcium, stents or metallic implants commonly degrade image clarity. These challenges have been partially overcome by using beta-blocker premedication, gating optimizations to reduce motion artifacts and post-processing software that refine the image [82,85]. A major limitation of CT is the exposure to ionizing radiation, which is of particular concern in younger patients or in those requiring repeated imaging, despite significant technological advances in reducing the radiation dose. Furthermore, use of iodinated contrast agents carries risks of allergic reactions and contrast-induced nephropathy, that requires caution in patients with renal insufficiency [81,86]. Finally, cardiac CT predominantly provides anatomical information and its role in functional assessment of ischemia and microvascular disease remains limited. The addition of CT-derived fractional flow reserve or myocardial perfusion imaging improves diagnostic accuracy but still cannot replace invasive coronary function testing or advanced modalities such as PET or CMR for definitive diagnosis in many cases [5,79,87].

## 7. Future Perspective

### Artificial Intelligence and Radiomics in Plaque Analysis with CT

Recent advances in cardiac CT now extend beyond conventional evaluation of stenosis and plaque burden to include radiomics and AI-based quantitative analysis [88]. These approaches enable extraction of detailed imaging features that are not visually apparent, improving risk stratification, prediction of adverse cardiac events, and assessment of plaque progression [89,90]. Although still requiring large-scale validation, radiomics and AI-driven automated plaque analysis represent a promising and rapidly evolving direction for non-invasive evaluation of coronary atherosclerosis.

## 8. Conclusions

NOCAD encompasses a heterogeneous but clinically significant group of conditions often presenting with angina, ischemia, or myocardial infarction despite the absence of obstructive stenoses. Cardiac CT has become a key diagnostic tool, expanding its role from ruling out obstructive CAD to a detailed assessment of coronary anatomy, plaque characteristics, and myocardial integrity. Advanced CT techniques, including fractional flow reserve, perfusion imaging, and novel biomarkers such as pericoronary fat attenuation index, provide additional insights into disease mechanisms. Despite these advances, challenges remain—particularly limited resolution for microvascular disease, radiation exposure, and lack of standardization. Further large-scale studies are needed to confirm the prognostic value of CT-derived parameters. Overall, cardiac CT enables integrated evaluation and risk stratification in NOCAD, and ongoing advances in imaging, computational modeling, and AI are likely to strengthen its diagnostic and prognostic role.

## Figures and Tables

**Figure 1 jcm-15-00032-f001:**
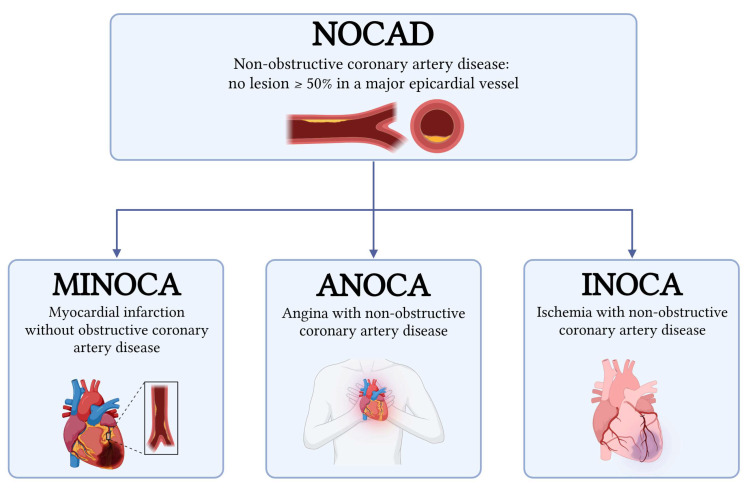
Definitions. Abbreviations: NOCAD = Non-Obstructive Coronary Artery disease. MINOCA = Myocardial Infarction with Non-Obstructive Coronary Arteries. ANOCA = Angina with Non-Obstructive Coronary Arteries. INOCA = Ischemia with Non-Obstructive Coronary Arteries.

**Figure 2 jcm-15-00032-f002:**
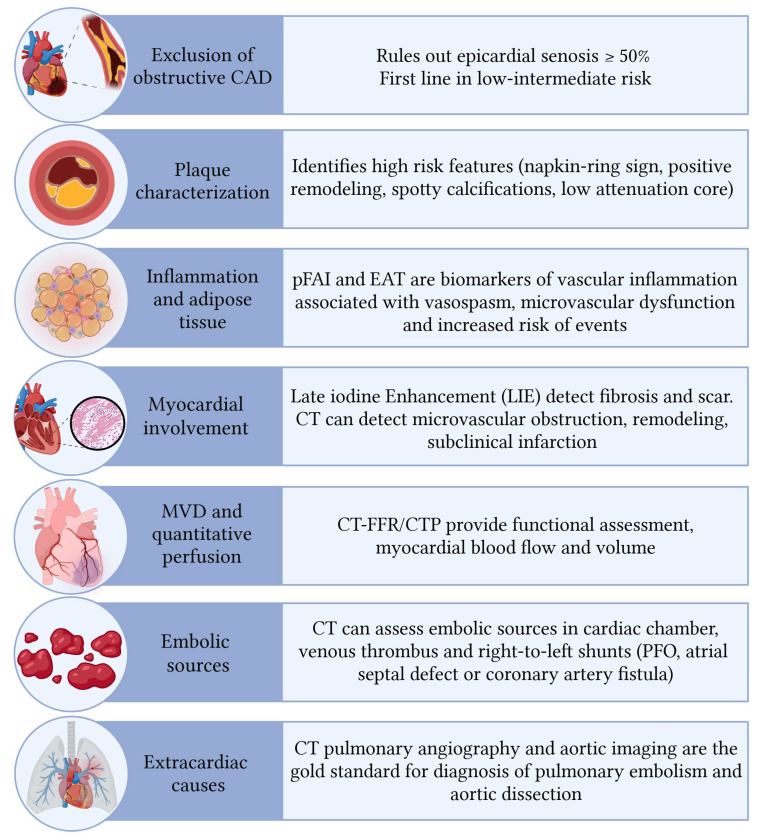
Application of Computed Tomography in the assessment of Non-Obstructive Coronary Artery Disease. Abbreviations: CAD = Coronary Artery Disease. pFAI = Pericoronary Fat Attenuation Index. EAT = epicardial adipose tissue. FFR-CT = Computed Tomography-Derived Fractional Flow Reserve. CTP = Computed Tomography Perfusion. CT = Computed Tomography. MVD = Micro Vascular Dysfunction. PFO = Patent Foramen Ovale.

**Figure 3 jcm-15-00032-f003:**
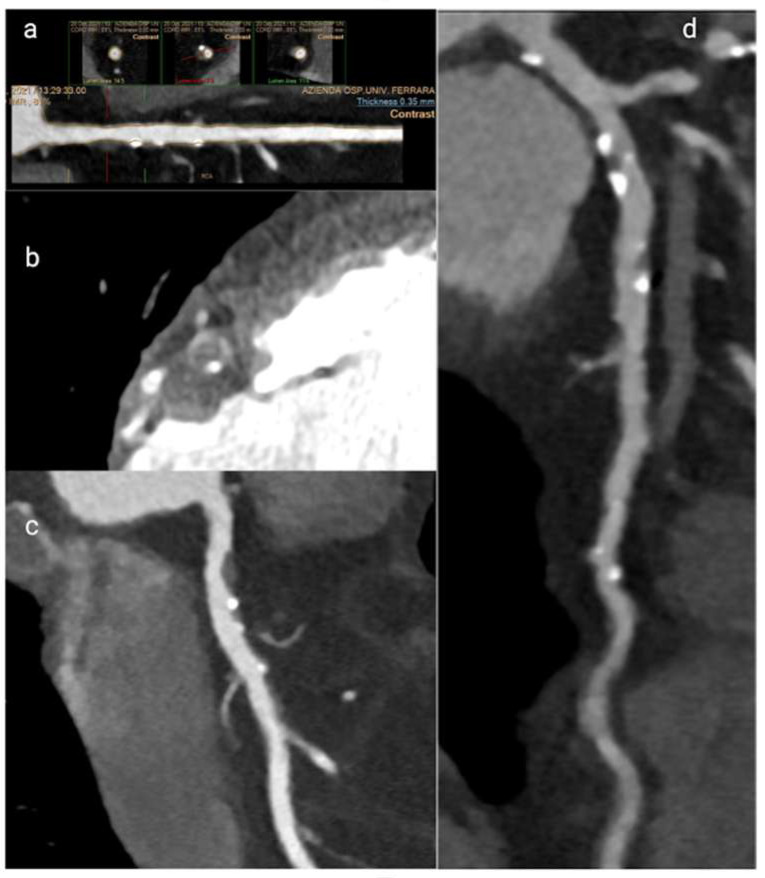
Computed Tomography images of the features that define a high-risk plaque. (**a**) positive remodelling (**b**) napkin-ring sign; (**c**) low-attenuation plaque; (**d**) spotty calcifications.

**Figure 4 jcm-15-00032-f004:**
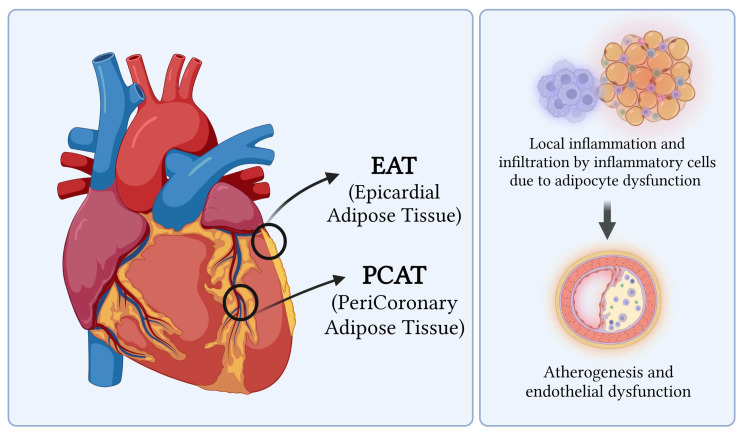
Epicardial and Pericoronary adipose tissue. Schematic representation of epicardial adipose tissue (EAT) and pericoronary adipose tissue (PCAT) and their role in coronary artery disease. Adipocyte dysfunction within EAT and PCAT promotes local inflammatory cell infiltration, which in turn contributes to perivascular inflammation, endothelial dysfunction, and atherogenesis in the adjacent coronary arteries.

**Figure 5 jcm-15-00032-f005:**
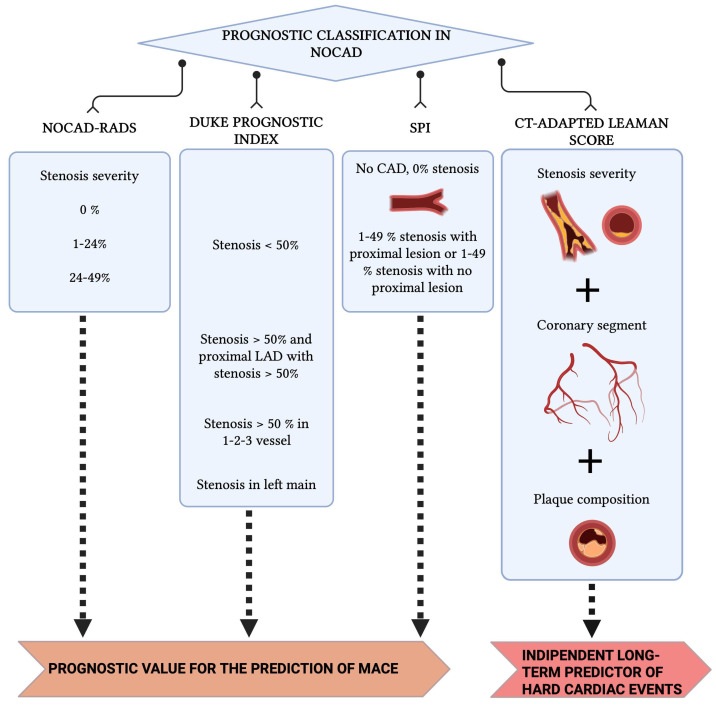
Prognostic classification derived from cardiac CT in NOCAD. Abbreviations: NOCAD = Non-Obstructive Coronary Artery Disease. CT = Computed Tomography. SPI = Stenosis Proximal Involvement.

**Table 1 jcm-15-00032-t001:** Main etiologies of MINOCA and ANOCA/INOCA. Abbreviations: MINOCA = Myocardial Infarction with Non-Obstructive Coronary Arteries. ANOCA = Angina with Non-Obstructive Coronary Arteries. INOCA = Ischemia with Non-Obstructive Coronary Arteries.

MINOCA	ANOCA/INOCA
Plaque disruption (rupture ulceration)	Microvascular dysfunction
Coronary vasospasm	Microvascular spasm
Coronary thrombosis and embolism	Epicardial spasm
Spontaneous coronary dissection	Endothelial dysfunction
Takotsubo syndrome	Myocardial bridging

**Table 2 jcm-15-00032-t002:** CT-derived biomarkers and their clinical implications in NOCAD.

CT-Derived Biomarker	Clinical Implications in NOCAD
*Pericoronary adipose tissue (PCAT) attenuation*	Reflects local coronary inflammation; associated with plaque vulnerability and adverse cardiovascular outcomes even in the absence of obstructive stenosis; may improve risk stratification in NOCAD patients.
*Epicardial adipose tissue (EAT) volume and density*	Marker of cardiometabolic risk and systemic inflammation; increased EAT burden is linked to coronary inflammation, endothelial dysfunction, and worse prognosis in NOCAD.
*High-risk plaque features (e.g., low-attenuation plaque, positive remodeling, napkin-ring sign)*	Identify vulnerable plaques that may underlie ischemia or acute coronary syndromes despite limited luminal narrowing; support intensified preventive therapy.
*Coronary plaque burden and composition*	Total plaque volume and non-calcified plaque components provide prognostic information beyond stenosis severity and help identify higher-risk NOCAD phenotypes.
*Coronary artery calcium (CAC) score*	Indicates overall atherosclerotic burden; useful for long-term risk assessment, although limited in capturing active inflammation or plaque vulnerability in NOCAD.
*CT-derived radiomics features*	Capture microstructural plaque and perivascular tissue characteristics not visible on visual assessment; may enhance prediction of plaque progression and adverse outcomes.
*AI-based automated plaque analysis*	Enables standardized, reproducible quantification of plaque and inflammatory biomarkers; facilitates integration of advanced metrics into clinical workflows and longitudinal monitoring.
*CT-derived functional indices (e.g., CT-FFR)*	Help assess the functional relevance of non-obstructive lesions and identify ischemia-related symptoms in NOCAD patients when anatomy alone is insufficient.

## Data Availability

No new data were created or analysed in this study. Data sharing is not applicable to this article.

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
