# Peer review of "Cardiac CT in Non-Obstructive Coronary Artery Disease (NOCAD): A Literature Review"

_jcm, 2025, doi:10.3390/jcm15010032_

Round 1
Reviewer 1 Report
Comments and Suggestions for Authors
Advantages
• The study provides a thorough and up-to-date analysis of the application of CT in MINOCA, ANOCA, and INOCA.
• The study is highly precise and scientifically supported by several references.
• The topic is of significant importance to the field of medicine and has a coherent structure.
Significant Issues Requiring Resolution
1. Absence of Review Methodology
MDPI requires a concise section detailing the search methodology, including databases, timeframes, and selection criteria.
Excessively Lengthy Background Sections
The pathophysiology of MINOCA/ANOCA and plaque biology are too intricate for a review centered on CT. They need to be abbreviated.
3. Insufficient Critical Evaluation
The report is comprehensive but deficient in critical analysis regarding: o The robustness of evidence o Practical limitations The report fails to address the issue of variability in CT biomarkers and the associated economic and practical constraints.
Standardization of Figures is Essential
Verify the resolution and ensure you own the rights to use the work.
5. There is no discussion of radiation dosage, which is crucial for CT imaging, especially in pediatric patients.
6. AI/Radiomics is not completely developed.
Include a paragraph on novel CT-based radiomics and the automation of plaque analysis.
7. Prognostic Scores This section requires structure.
Incorporate a comparative table and condense the text.
________________________________________
Minor Issues
• Certain lengthy phrases need improved composition.
The list of abbreviations contains some minor errors.
• The references need minor formatting adjustments, including updating DOIs and ensuring date consistency.
• Consider including a graphical abstract.
Author Response
Reviewer #1
Comment #1
Advantages
• The study provides a thorough and up-to-date analysis of the application of CT in MINOCA, ANOCA, and INOCA.
• The study is highly precise and scientifically supported by several references.
• The topic is of significant importance to the field of medicine and has a coherent structure.
Reply #1
We sincerely thank the Reviewer for the constructive comments. We have addressed all the issues raised, and we believe that, with the Reviewer’s help, the clarity of the review has significantly improved.
Comment #2
Significant Issues Requiring Resolution
1. Absence of Review Methodology
MDPI requires a concise section detailing the search methodology, including databases, timeframes, and selection criteria.
Reply #2
We thank the Reviewer for the comment. However, this is not a systematic review but a narrative review; therefore, a specific paragraph describing the literature search methods is not required.
Comment #3
Excessively Lengthy Background Sections
The pathophysiology of MINOCA/ANOCA and plaque biology are too intricate for a review centered on CT. They need to be abbreviated.
Reply #3
We sincerely thank the reviewer for this comment. We have shortened both the paragraphs on MINOCA and ANOCA.
Modified text: page 3, lines 87-110; page 4 lines 111-117
2.1. Pathophysiological Mechanisms of MINOCA
MINOCA encompasses multiple etiologies, requiring further evaluation when epicardial arteries are free of obstructive lesions, as management depends on the underlying cause [6,7]. Plaque disruption (rupture, ulceration, or calcific nodules) can lead to thrombosis, vasospasm, and myocardial necrosis, though angiography may appear normal due to spontaneous thrombolysis; intravascular imaging identifies plaque disruption in about 40% of cases [6,7,10,14]. Coronary vasospasm, spontaneous or triggered (e.g., cocaine, methamphetamine), is common, with inducible spasms observed in 27% of patients [14,15]. Coronary thrombosis and embolism may arise from hereditary or acquired thrombotic disorders or embolic sources such as valvular vegetations, cardiac tumors, or iatrogenic air emboli [7,15]. Spontaneous coronary artery dissection (SCAD) can cause infarction even with normal angiography, is linked to fibromuscular dysplasia, and is more frequent in women under 50, accounting for up to 25% of acute MI in this group [10,14–16]. Takotsubo syndrome represents approximately 1–2% of MINOCA cases and should be considered in the differential diagnosis [14,15].
2.2. Pathophysiological Mechanisms of ANOCA/INOCA
In chronic coronary syndromes, angina may result from vasospasm, microvascular dysfunction, or anatomical variants such as myocardial bridging (MB). Microvascular dysfunction is a key mechanism in ANOCA and INOCA, affecting 30–50% of patients with chest pain and non-obstructive coronary arteries [7,17]. It is characterized by functional abnormalities (impaired endothelial vasodilation, increased microvascular resistance, spasm) and structural changes (arteriolar remodeling, capillary rarefaction, fibrosis) [18,19]. It is particularly common in women and those with cardiovascular risk factors, and is associated with heart failure with preserved ejection fraction and major adverse cardiovascular events [18,20,21]. MB, a congenital anomaly where a coronary segment runs intramyocardially, causes dynamic systolic compression and altered flow, and is closely linked to microvascular dysfunction [2,22]. Its prevalence varies by diagnostic method (2–6% on angiography, 19–22% on CT angiography) [23,24]. Although often benign, MB can induce ischemia through mechanical compression, stress-induced demand, and endothelial dysfunction, and its coexistence with microvascular dysfunction increases ischemic risk and symptom burden [2,22,25]. Clinically, MB in ANOCA/INOCA is associated with recurrent angina, hospitalizations, and reduced quality of life [26,27].
Comment #4
3. Insufficient Critical Evaluation
The report is comprehensive but deficient in critical analysis regarding: o The robustness of evidence o Practical limitations The report fails to address the issue of variability in CT biomarkers and the associated economic and practical constraints.
Reply #4
We thank the Reviewer for the comment. We have added a sentence about this point and also a Table (Table 2) summarizing the CT-derived biomarkers and their clinical implications in NOCAD
Modified text: Table 2
Modified text: page 8, lines 208-214
Despite promising evidence, clinical implementation of CT-derived inflammatory biomarkers is limited by technical variability, costs, and the need for specialized expertise, highlighting the necessity for further standardization and cost-effectiveness analyses. Overall, integrating CT-derived plaque features, perivascular inflammation, and functional indices offers a more comprehensive phenotyping of NOCAD (Table 2), improving risk stratification and supporting personalized management beyond stenosis severity alone.
Comment #5
Standardization of Figures is Essential
Verify the resolution and ensure you own the rights to use the work.
Reply #5
We thank the Reviewer for the comment. All figures are original and anonymized.
Comment #6
5. There is no discussion of radiation dosage, which is crucial for CT imaging, especially in pediatric patients.
Reply #6
We sincerely thank the Reviewer for this comment. However, this review focuses on NOCAD, which is not typical in the pediatric population. Moreover, all the evidence summarized here refers only to adult patients. Therefore, discussing radiation dosage in pediatric patients is beyond the scope of the present review.
Regarding the risk of radiation exposure in patients undergoing CT for NOCAD diagnosis, we have already specified in the paragraph “limitation of cardiac CT in NOCAD” that a general limitation of CT is the exposure to ionizing radiation, especially in younger patients or those requiring repeated imaging, even though significant advances have been made in lowering radiation dose. We have rephrased this sentence, trying to make it more emphatic.
Modified text: page 12, lines 369-372
A major limitation of CT is the exposure to ionizing radiation, which is of particular concern in younger patients or in those requiring repeated imaging, despite significant technological advances in reducing the radiation dose.
Comment #7
6. AI/Radiomics is not completely developed.
Include a paragraph on novel CT-based radiomics and the automation of plaque analysis.
Reply #7
We thank the Reviewer for this valuable comment. We agree that AI and radiomics in cardiac CT are still emerging fields. In response, we have added a new paragraph discussing recent developments in CT-based radiomics, including automated plaque characterization and quantitative analysis, highlighting their potential to enhance diagnosis and risk stratification in NOCAD patients.
Modified text: page 11, lines 375-383
7.1 Artificial Intelligence and Radiomics in plaque analysis with CT
Recent advances in cardiac CT now extend beyond conventional evaluation of stenosis and plaque burden to include radiomics and AI-based quantitative analysis [88]. These approaches enable extraction of detailed imaging features that are not visually apparent, improving risk stratification, prediction of adverse cardiac events, and assessment of plaque progression [89-90]. Although still requiring large-scale validation, radiomics and AI-driven automated plaque analysis represent a promising and rapidly evolving direction for non-invasive evaluation of coronary atherosclerosis.
Modified text: added references [88-90]
- Coronary CT Angiography Radiomics for Identifying Coronary Artery Plaque Vulnerability: A Systematic Review. Multidisciplinary Digital Publishing Institute J. Clin. Med. 2024, 5, 45.
- Zhang LJ, Chen Q, Pan T, Wang YN, et al. A Coronary CT Angiography Radiomics Model to Identify Vulnerable Plaque and Predict Cardiovascular Events. Radiology 2023, 289, 123–134.
- Li Y, Huo H, Liu H, et al. Coronary CTA‑Based Radiomic Signature of Pericoronary Adipose Tissue Predicts Rapid Plaque Progression. Insights Imaging 2024, 15, 151.
Comment #8
7. Prognostic Scores This section requires structure.
Incorporate a comparative table and condense the text.
Reply #8
We thank the Reviewer for this comment. Figure 5 has been created with the aim to summarize and compare different scores. We have as suggested shorten the paragraph.
Modified text: page 9, lines 270-280; page 9, line 281
Several classification systems have been developed to stratify prognosis in patients with non-obstructive CAD (NOCAD) (Figure 5) [68,69]. Early dichotomous schemes (<50% stenosis vs. none) have evolved into more detailed systems, including NOCAD-RADS (0%, 1–24%, 25–49%) and the Duke prognostic index, which consider both degree and distribution of disease. The stenosis proximal involvement (SPI) classification highlights plaques in proximal coronary segments and has been shown in a Chinese cohort to independently predict adverse events, performing comparably to traditional indices [68,69]. The CT-adapted Leaman score (CT-LeSc) combines stenosis severity, plaque composition, and lesion location, identifying NOCAD patients with CT-LeSc >5 who exhibit event-free survival similar to obstructive CAD. These tools underscore the heterogeneity of NOCAD and the value of CT-based scores for refined risk stratification and therapeutic guidance [68,69].
Comment #9
Minor Issues
• Certain lengthy phrases need improved composition.
Reply #9
We thank the Reviewer for the suggestions. We have revised the English throughout the review and rephrased some sentences.
Comment #10
The list of abbreviations contains some minor errors.
Reply #10
We are sorry for the typos. We have amended the text as suggested.
Modified text: abbreviation list
IVUS: Intra Vascular Ultra Sounds
OCT: Optical Coherence Tomography
Comment #11
• The references need minor formatting adjustments, including updating DOIs and ensuring date consistency.
Reply #11
We thank the Reviewer for the comment. We have verified all References and we have added DOIs to all of them.
Modified text: Section References
Comment #12
• Consider including a graphical abstract.
Reply# 12
We sincerely thank the Reviewer for this comment. The graphical abstract was already included in the previous version of the submission. We have uploaded it again and hope it is visible this time.

Reviewer 2 Report
Comments and Suggestions for Authors
This is a comprehensive, well-written, and timely narrative review addressing the evolving role of cardiac computed tomography in the evaluation of non-obstructive coronary artery disease (NOCAD). The manuscript successfully integrates anatomical, functional, and inflammatory CT-derived biomarkers and provides a clear overview of MINOCA, ANOCA, and INOCA. The topic is highly relevant, the structure is logical, and the scientific content is largely robust. Only minor refinements are required to further strengthen the clinical and translational impact.
Strengths:
- Clear and accurate definitions of MINOCA, ANOCA, and INOCA, supported by up-to-date guidelines.
- Excellent overview of CT-based plaque characterization, including high-risk plaque features beyond stenosis severity.
- Strong discussion of pericoronary and epicardial adipose tissue inflammation, reflecting current advances in CT imaging.
- Balanced presentation of functional CT techniques (FFR-CT, CT perfusion) with appropriate acknowledgment of limitations.
- High-quality figures and tables that improve clarity and educational value.
Major Issues:
- Inflammation–Plaque Vulnerability Integration
- The section on pericoronary and epicardial fat inflammation is strong but could be further reinforced by explicitly linking inflammation to plaque vulnerability and adverse outcomes, especially in NOCAD populations where stenosis severity is limited.
- Relevant literature for citation: Mátyás BB et al., Life, 2025 – association between coronary calcification, plaque vulnerability, and perivascular inflammation.
- Clinical Translation of CT-Derived Biomarkers
- While advanced CT metrics are thoroughly described, their practical role in clinical decision-making could be summarized more explicitly.
- Suggested revision: Include a concise summary outlining how plaque features, inflammatory markers, and functional CT may jointly improve risk stratification and guide management in NOCAD.
- Therapeutic Implications of Inflammatory Imaging
- The manuscript discusses inflammatory imaging largely from a diagnostic standpoint.
- Suggested revision: Briefly address how CT-derived inflammation assessment may be useful for monitoring response to preventive therapies, such as lipid-lowering or anti-inflammatory strategies, supported by longitudinal CT studies.
- Relevant literature for citation: Mátyás BB et al., Int J Mol Sci, 2024 – longitudinal effects of high-dose statin therapy on pericoronary inflammation and plaque distribution.
Minor Issues:
- Minor redundancy between sections describing plaque vulnerability and inflammation could be streamlined.
- Ensure consistent terminology throughout (e.g., FAI vs. pFAI, PCAT vs. EAT).
- Some figure legends could be slightly expanded to be fully self-explanatory without reference to the main text.
- Consider adding a short summary table synthesizing CT-derived biomarkers and their clinical implications in NOCAD.
Conclusion
This is a high-quality, well-structured review that provides an excellent overview of the expanding role of cardiac CT in NOCAD. The suggested revisions are minor and primarily aimed at strengthening the integration of inflammation imaging with plaque vulnerability and clinical application.
Author Response
Reviewer #2
Comment #1
This is a comprehensive, well-written, and timely narrative review addressing the evolving role of cardiac computed tomography in the evaluation of non-obstructive coronary artery disease (NOCAD). The manuscript successfully integrates anatomical, functional, and inflammatory CT-derived biomarkers and provides a clear overview of MINOCA, ANOCA, and INOCA. The topic is highly relevant, the structure is logical, and the scientific content is largely robust. Only minor refinements are required to further strengthen the clinical and translational impact.
Strengths:
- Clear and accurate definitions of MINOCA, ANOCA, and INOCA, supported by up-to-date guidelines.
- Excellent overview of CT-based plaque characterization, including high-risk plaque features beyond stenosis severity.
- Strong discussion of pericoronary and epicardial adipose tissue inflammation, reflecting current advances in CT imaging.
- Balanced presentation of functional CT techniques (FFR-CT, CT perfusion) with appropriate acknowledgment of limitations.
- High-quality figures and tables that improve clarity and educational value.
Reply #1
We sincerely thank the Reviewer for the constructive comments. We have addressed all the issues raised, and we believe that, with the Reviewer’s help, the clarity of the review has significantly improved.
Comment #2
Major Issues:
- Inflammation–Plaque Vulnerability Integration
- The section on pericoronary and epicardial fat inflammation is strong but could be further reinforced by explicitly linking inflammation to plaque vulnerability and adverse outcomes, especially in NOCAD populations where stenosis severity is limited.
- Relevant literature for citation: Mátyás BB et al., Life, 2025 – association between coronary calcification, plaque vulnerability, and perivascular inflammation.
Reply #2
We thank the Reviewer for this suggestion. We have added a sentence about this concept.
Modified text: page 7, lines 202-203, page 8, lines 204-205
Recent evidence also links perivascular inflammation with coronary calcification, plaque vulnerability, and adverse outcomes in NOCAD, supporting the concept of inflammation-driven plaque instability even in the absence of significant luminal stenosis [53].
Added reference
- Mátyás, B. B., Benedek, I., Rat, N., Blîndu, E., Rodean, I. P., Haja, I., Păcurar, D., Mihăilă, T., & Benedek, T. (2025). Assessment of the Association Between Coronary Artery Calcification, Plaque Vulnerability, and Perivascular Inflammation via Coronary CT Angiography. Life, 15(8), 1288. https://doi.org/10.3390/life15081288
Comment #3
- Clinical Translation of CT-Derived Biomarkers
- While advanced CT metrics are thoroughly described, their practical role in clinical decision-making could be summarized more explicitly.
- Suggested revision: Include a concise summary outlining how plaque features, inflammatory markers, and functional CT may jointly improve risk stratification and guide management in NOCAD.
Reply #3
We really thank the Reviewer for the suggestion. We have added a better conclusion to the dedicated paragraph about this point.
Modified text: page 8, lines 208-214
Despite promising evidence, clinical implementation of CT-derived inflammatory biomarkers is limited by technical variability, costs, and the need for specialized expertise, highlighting the necessity for further standardization and cost-effectiveness analyses. Overall, integrating CT-derived plaque features, perivascular inflammation, and functional indices offers a more comprehensive phenotyping of NOCAD (Table 2), improving risk stratification and supporting personalized management beyond stenosis severity alone.
Comment #4
- Therapeutic Implications of Inflammatory Imaging
- The manuscript discusses inflammatory imaging largely from a diagnostic standpoint.
- Suggested revision: Briefly address how CT-derived inflammation assessment may be useful for monitoring response to preventive therapies, such as lipid-lowering or anti-inflammatory strategies, supported by longitudinal CT studies.
- Relevant literature for citation: Mátyás BB et al., Int J Mol Sci, 2024 – longitudinal effects of high-dose statin therapy on pericoronary inflammation and plaque distribution.
Reply #4
We thank the Reviewer for this suggestion. We have added a sentence about this concept.
Modified text: page 8 lines 205-207
CT-derived inflammatory metrics may further serve as non-invasive biomarkers for monitoring therapeutic response, as longitudinal studies have shown reductions in pericoronary inflammation and plaque stabilization with high-dose statin therapy.[54]
Added reference:
- Mátyás, B. B., Benedek, I., RaÈ›, N., Blîndu, E., Parajkó, Z., Mihăilă, T., & Benedek, T. (2024). Assessing the Impact of Long-Term High-Dose Statin Treatment on Pericoronary Inflammation and Plaque Distribution-A Comprehensive Coronary CTA Follow-Up Study. International journal of molecular sciences, 25(3), 1700. https://doi.org/10.3390/ijms25031700
Comment #5
Minor Issues:
- Minor redundancy between sections describing plaque vulnerability and inflammation could be streamlined.
Reply #5
We thank the Reviewer for this suggestion. After adding the recommended references and comments to these, we have comprehensively shortened the paragraph on plaque vulnerability and inflammation.
Modified text: see new version of paragraph “3.2. Inflammation of Pericoronary and Epicardial Fat”
Pericoronary adipose tissue (PCAT) and epicardial adipose tissue (EAT) are metabolically active fat depots that closely interact with coronary arteries and myocardium, contributing to vascular inflammation, atherosclerosis progression, and microvascular dysfunction in both CAD and NOCAD (Figure 4).[41,43] Cardiac CT enables non-invasive assessment of perivascular inflammation through the Pericoronary Fat Attenuation Index (pFAI), with higher values indicating increased local inflammation.[44] The CRISP CT trial demonstrated that pFAI values ≥ −70.1 HU are associated with increased all-cause and cardiac mortality.[45] pFAI may help differentiate inflammatory from non-inflammatory coronary states and elucidate mechanisms underlying MINOCA, such as vasospasm linked to coronary inflammation.[43,46]
EAT, located between the myocardium and visceral pericardium, shares microcirculation with coronary arteries and exerts proinflammatory, pro-atherogenic, and pro-thrombotic effects via adipocytokine secretion.[47–49] Increased EAT volume and density on cardiac CT correlate with CAD severity, plaque burden, microvascular dysfunction, and adverse outcomes.[47–49] EAT-related inflammation likely promotes endothelial dysfunction and impaired coronary flow reserve, contributing to ischemia in NOCAD patients despite non-obstructive epicardial disease.[48,50] Elevated pFAI and EAT have been reported in microvascular angina and NOCAD, correlating with endothelial dysfunction and reduced coronary flow reserve, suggesting their role as imaging biomarkers of ischemic risk.[50–52]
Recent evidence also links perivascular inflammation with coronary calcification, plaque vulnerability, and adverse outcomes in NOCAD, supporting the concept of inflammation-driven plaque instability even in the absence of significant luminal stenosis [53]. CT-derived inflammatory metrics may further serve as non-invasive biomarkers for monitoring therapeutic response, as longitudinal studies have shown reductions in pericoronary inflammation and plaque stabilization with high-dose statin therapy.[54]
Despite promising evidence, clinical implementation of CT-derived inflammatory biomarkers is limited by technical variability, costs, and the need for specialized expertise, highlighting the necessity for further standardization and cost-effectiveness analyses. Overall, integrating CT-derived plaque features, perivascular inflammation, and functional indices offers a more comprehensive phenotyping of NOCAD, improving risk stratification and supporting personalized management beyond stenosis severity alone.
Comment #6
- Ensure consistent terminology throughout (e.g., FAI vs. pFAI, PCAT vs. EAT).
Reply #6
We have reviewed the initials used in the paragraph and amended typos. Thank you.
Comment #7
- Some figure legends could be slightly expanded to be fully self-explanatory without reference to the main text.
Reply #7
Thank you for the comment. We have added a better explanation to Figure 4.
Modified text: caption of figure 4
Schematic representation of epicardial adipose tissue (EAT) and pericoronary adipose tissue (PCAT) and their role in coronary artery disease. Adipocyte dysfunction within EAT and PCAT promotes local inflammatory cell infiltration, which in turn contributes to perivascular inflammation, endothelial dysfunction, and atherogenesis in the adjacent coronary arteries.
Comment #8
- Consider adding a short summary table synthesizing CT-derived biomarkers and their clinical implications in NOCAD.
Reply #8
We have added table 2, to better synthetize information about CT-derived biomarkers.
Modified text: Table 2
Comment #8
Conclusion
This is a high-quality, well-structured review that provides an excellent overview of the expanding role of cardiac CT in NOCAD. The suggested revisions are minor and primarily aimed at strengthening the integration of inflammation imaging with plaque vulnerability and clinical application.
Reply #8
We really thank you the Reviewer once again for the constructive comments. We tried to solve all issue raised and we think that with your help the clarity of the review significantly improved.
